# Bayesian Analysis of Femtosecond Pump-Probe Photoelectron-Photoion Coincidence Spectra with Fluctuating Laser Intensities

**DOI:** 10.3390/e21010093

**Published:** 2019-01-19

**Authors:** Pascal Heim, Michael Rumetshofer, Sascha Ranftl, Bernhard Thaler, Wolfgang E. Ernst, Markus Koch, Wolfgang von der Linden

**Affiliations:** 1Institute of Experimental Physics, Graz University of Technology, 8010 Graz, Austria; 2Institute of Theoretical and Computational Physics, Graz University of Technology, 8010 Graz, Austria

**Keywords:** photoelectron-photoion coincidence, PEPICO, femtosecond pump-probe spectroscopy, ultrafast molecular dynamics, Bayesian data analysis

## Abstract

This paper employs Bayesian probability theory for analyzing data generated in femtosecond pump-probe photoelectron-photoion coincidence (PEPICO) experiments. These experiments allow investigating ultrafast dynamical processes in photoexcited molecules. Bayesian probability theory is consistently applied to data analysis problems occurring in these types of experiments such as background subtraction and false coincidences. We previously demonstrated that the Bayesian formalism has many advantages, amongst which are compensation of false coincidences, no overestimation of pump-only contributions, significantly increased signal-to-noise ratio, and applicability to any experimental situation and noise statistics. Most importantly, by accounting for false coincidences, our approach allows running experiments at higher ionization rates, resulting in an appreciable reduction of data acquisition times. In addition to our previous paper, we include fluctuating laser intensities, of which the straightforward implementation highlights yet another advantage of the Bayesian formalism. Our method is thoroughly scrutinized by challenging mock data, where we find a minor impact of laser fluctuations on false coincidences, yet a noteworthy influence on background subtraction. We apply our algorithm to data obtained in experiments and discuss the impact of laser fluctuations on the data analysis.

## 1. Introduction

Coincidence measurements are a widely-used and powerful experimental technique in physics and chemistry. Photoelectron-photoion coincidence (PEPICO) spectroscopy utilizes not only information obtained from the detection of electrons and ions, but also the fact that they stem from the very same ionization event [1,2,3,4,5,6]. Frequently used in photoionization studies of gas phase molecules or clusters, this technique allows for conclusions about the ionization process such as the disentanglement of competing intramolecular relaxation channels [4,7,8,9] or multiple species [10], a depth of insight that cannot be achieved without assigning electrons to the ions from which they originate. Thus, the success of PEPICO is based on these recordings of pairs being unambiguous, energy-resolved for electrons, and mass-resolved for the cations. Yet, the correct pairwise assignment (true coincidence) may be affected by certain experimental conditions: If a laser pulse triggers a number of simultaneous ionization events arising from different neutral molecules, the assignment of correlated electron-ion pairs is impaired and causes so-called false coincidences [11]. The three possible events are: (1) Not exactly one electron and one ion are detected; in this case the event is rejected. (2) One electron and one ion are detected, which can originate from the same molecule (true coincidence) or (3) an electron and ion can originate from different molecules (false coincidence). Let this be illustrated using the example of exactly two ionization events. If two electrons and/or two ions are detected, the measurement would simply be discarded since no unambiguous assignments could be made. Yet, with imperfect detectors, there is a non-negligible probability of the following event: The electron from Molecule 1 is detected, the electron from Molecule 2 is not detected; Cation 1 is not detected; Cation 2 is detected. Hence, the experimentalist sees a false coincidence, where Electron 1 is wrongly assigned to Cation 2. Obviously, false coincidences only arise if both detectors are not perfect, i.e., the detection probabilities are less than unity, and are thus to some extent present in any such experiment. An easy way out is to work with low ionization rates, for the price of either a bad signal-to-noise ratio or time-consuming measurements. In principle, detector noise or ionization events not caused by the laser pulse might also lead to false coincidences, but, for the situation at hand, are sufficiently low to be neglected.

Time-resolved studies are typically carried out as pump-probe experiments [10,12] as depicted in Figure 1. Excitation by a laser pulse, commonly referred to as the pump pulse, triggers dynamical processes in the molecule after which a time-delayed second laser pulse, commonly referred to as the probe pulse, ionizes the molecule. The transient change of photo-electron and -ion signals associated with the excited states, as a function of the time-delay, provides insight into the underlying processes. Unfortunately, pump and/or probe pulses on their own can ionize the molecule as well, leading to signals that are referred to as pump-only and probe-only further on. This background signal is superimposed on the excited state signal, and in many experimental situations, the pump-only and/or the probe-only signals significantly contribute to the pump-probe signal, e.g., if multiphoton transitions are applied for pumping or probing, or if high photon energies are used for probing [13]. In order to extract the excited state transients, the pump-only and/or the probe-only signals are measured separately and usually subtracted from the pump-probe signals, obviously resulting in increased noise if the background and pump-probe spectra energetically overlap with each other. The application of the Bayesian formalism to background subtraction alone was already presented for astrophysical applications [14,15] and for photo-induced X-ray emission spectroscopy (PIXE) [16,17,18,19,20,21]. Yet, here, it has to be considered that the pump-only, the probe-only, and the pump-probe measurements have different ionization rates. Since the statistics of the coincidences depend on the ionization rates, the statistics of sole pump-only and probe-only measurements differ from those in the pump-probe measurement, and simple subtraction turns out not to be an unbiased estimator any longer [22].

Although it is feasible to distinguish between true and false coincidences by pure experimental finesse, such as in cold target recoil ion momentum spectroscopy (COLTRIMS) [23,24], it demands quite a technical and financial effort and is entirely impossible for time-of-flight detection, as used in the presented experiment. Covariance mapping, which is based on the calculation of the covariance for the photoelectron and mass spectra measured with each laser shot [25,26,27], does not guarantee that the reconstructed spectrum is positive, is restricted to Poisson processes, and leads to systematic deviations in other scenarios [26,27]. Further limitations are outlined in [25].

We recently presented a Bayesian approach to PEPICO, which treats both coincidences and background subtraction on the same footing [22,29]. In this work, we extend our theory to the prevalent experimental situation of unstable laser intensity, i.e., ionization rates fluctuating from pulse to pulse. We provide our software, including introductory examples, at https://github.com/fslab-tugraz/PEPICOBayes/. The experiment described in Section 2 is treated with the Bayesian formalism developed in Section 3. It will be tested by some challenging mock data in Section 4 and applied to real experimental data in Section 5.

## 2. Experiment

The analyzed experiment is of the type depicted in Figure 1 and described in detail in previous publications [7,28]. To apply our method, we choose our specimen and excitation-ionization-scheme according to Figure 2, since we expect the effects described above to be of particular importance in this scenario. Acetone molecules are excited by a three-photon transition to high-lying Rydberg states and ionized in the extraction region of a time-of-flight spectrometer, which measures both the electron kinetic energies and the ion masses [4,7]. The electron and ion flight times are then analyzed by a coincidence algorithm to produce photoelectron spectra corresponding to either an intact parent acetone ion or a fragmented acetyl ion. The concept and the necessity of time-resolved PEPICO (TR-PEPICO) in this context are further elucidated in Animations A1 and A2 in the Appendix A.

The addressed excited state lies energetically close to the ionization continuum, resulting in a certain probability of four-photon ionization from the ground state caused by the pump pulse alone (measurement α and Channel 1), leading to a background signal. In a separate pump-probe measurement (measurement β), the pump process is the same as in the pump-only case. The population is generated in the excited states, which in turn is ionized by a time-delayed probe pulse. Due to the low laser intensity of the probe pulse, ground state molecules are not ionized by the probe pulse alone, i.e., there is no probe-only background. Consequently, the measured pump-probe spectrum consists of pump-only ionization events (Channel 1) and pump-probe ionization events (Channel 2). Cations, produced in both channels, dependent on the ionization path, can either be stable and detected as parent ions or undergo fragmentation into neutral and ionic fragments. Coincidence detection of electrons and ions allows obtaining separate electron spectra for each ion, i.e., parent and fragment. The excited state of the molecule at the moment of ionization is identified by the measured electron kinetic energy in combination with the energy of the ionizing photon and knowledge of the vertical ionization energy of the excited state. In addition to the information of species and electronic state that is ionized, the related ion mass of the PEPICO spectrum provides insight into the fragmentation behavior. For example, the assignment of the photoelectron kinetic energy to an excited electronic state of the unfragmented molecule and coincidence detection of an ion fragment show that the molecule was intact at the moment of ionization and that fragmentation must have occurred afterwards. Moreover, the population in the excited state can decay to energetically lower states quite quickly, e.g., on a femtosecond timescale [4,7,28]. It is due to this decay that the Channel 2 signal can become significantly smaller than the Channel 1 background, in particular for long delay times, causing a poor signal-to-noise ratio.

## 3. Bayesian Data Analysis

### 3.1. Preliminary Considerations

We now introduce our notation and develop the Bayesian algorithm for analyzing the data generated in the experiment described in Section 1 and Section 2. We consider the following standard setup consisting of two experiments on the same target: pump-only and pump-probe, denoted by α and β, respectively. Each experiment consists of $N_{p}$ measurements. A measurement of the α experiment is performed with exactly one pump pulse, while a measurement of the β experiment comprises exactly one pump pulse and one probe pulse. During one measurement, two types of elementary coincidence events are detected, either a molecule is ionized from its ground state (referred to as Channel 1) or from its excited state (Channel 2). The latter is only possible in a pump-probe measurement (β). We assume that the number $m_{j}$ of ionization events in channel j∈{1,2} is Poisson distributed with some mean ionization rate $\lambda_{j}$. For the sake of readability, we suppress the index *j* in the following considerations. Furthermore, we assume that in each experiment, characterized by a defined delay time between pump and probe pulse, λ is independent of the occupation of the states, which means that we neglect population depletion effects. We additionally presume the laser intensity, and thus the ionization rate λ, to fluctuate. Mikosch and Patchkovskii [26,27] proposed to describe λ with a Gaussian Probability Density Function (PDF). We rather resort to a Γ-distribution of the latter instead, since: (1) the assignment of a Gaussian PDF for λ is inconsistent with the fact that λ≥0, while the Γ-distribution includes this constraint naturally; (2) it turns out to be quite convenient mathematically, while the effect of this choice on the result is deemed negligible, which will become apparent later on. We parametrize the Γ-distribution, $p_{\Gamma }\left(\lambda |\underline{\lambda },\sigma \right)$, with their expectation value $\underline{\lambda }$ and variance $\sigma^{2}$. All results match the findings of our previous paper [22] in the limit σ→0.

In one elementary event, the involved molecule can have mass $M_{\mu }$ and the emitted electron energy $E_{\nu }$. For brevity, we will refer to this particular event as (μν). The ion masses and the electron energies are discretized, μ,ν∈N, due to the finite resolution of the time-of-flight spectrometer. We will also use the symbol ρ∈{α,β}, if we refer to the measurements/experiments α or β, the symbol *j* for Channel 1 or 2, and *x* for the combination of both sets, i.e., x∈{1,2,α,β}. Given that an elementary event happens in channel j∈{1,2}, the probability that it corresponds to (μν) is denoted by:$$\begin{array}{cc} q_{\mu \nu }^{\left(j\right)} & =P(M_{\mu },E_{\nu }|\mathrm{one}\;\mathrm{elementary}\;\mathrm{event}\;\mathrm{in}\;\mathrm{channel}\;j,I)\; \end{array}$$
where I denotes the conditional complex. The probabilities are properly normalized:(1)$$\begin{array}{c} \sum_{\mu \nu }q_{\mu \nu }^{\left(j\right)}=1\;\;\;\;\forall j\in \left\{1,2\right\}\;. \end{array}$$
In the pump-only measurement (α), all molecules are in their respective ground state; therefore, only Channel 1 is allowed and: $$\begin{array}{c} q_{\mu \nu }^{\left(\alpha \right)}=q_{\mu \nu }^{\left(1\right)}\;. \end{array}$$

We now introduce the spectrum:$$\begin{array}{c} {\tilde{q}}_{\mu \nu }^{\left(1\right)}=P\left(E_{\nu },M_{\mu }|\mathrm{SC},q^{\left(1\right)},\pi ,I\right) \end{array}$$
with a subtle distinction of $q^{\left(1\right)}$ in the explicit propositions. $q^{\left(1\right)}$ is conditioned on (μν) being a true single coincidence, whereas ${\tilde{q}}_{\mu \nu }^{\left(1\right)}$ is conditioned on (μν) being a single coincidence, either true or false, and hence also on π. The latter is what is actually observed in a pump-only PEPICO measurement, while the former is what is desired. It will become apparent in the following how this additional condition distorts our statistics. We summarize all unknown parameters in the variable $\pi :=\{{\underline{\lambda }}_{1},\sigma_{1},{\underline{\lambda }}_{2},\sigma_{2},\xi_{i},\xi_{e}\}$. ${\underline{\lambda }}_{1}$, $\sigma_{1}$, ${\underline{\lambda }}_{2}$, and $\sigma_{2}$ describe the fluctuations of $\lambda_{1}$ and $\lambda_{2}$ according to Γ-distributions, and $\xi_{i}$ and $\xi_{e}$ are the detection probabilities of ions and electrons, respectively. The probability of detecting an electron with energy $E_{\nu }$ and an ion with mass $M_{\mu }$ in a single coincidence measurement is:(2)$$\begin{array}{cc} P(E_{\nu },M_{\mu },\mathrm{SC}|q^{\left(1\right)},\pi ,I) & =\int_{0}^{\infty }d\lambda_{1}\;P\left(E_{\nu },M_{\mu },\mathrm{SC}|q^{\left(1\right)},\lambda_{1},I\right)p_{\Gamma }\left(\lambda_{1}|{\underline{\lambda }}_{1},\sigma_{1}\right)\\ & =\xi_{e}\xi_{i}\left(q_{\mu \nu }^{\left(1\right)}\langle \lambda_{1}^{1}\rangle +{\bar{\xi }}_{e}{\bar{\xi }}_{i}q_{\mu .}^{\left(1\right)}q_{.\nu }^{\left(1\right)}\langle \lambda_{1}^{2}\rangle \right)\;, \end{array}$$
with the marginals $q_{.\nu }^{\left(j\right)}=\sum_{\mu }q_{\mu \nu }^{\left(j\right)}$ and $q_{\mu .}^{\left(j\right)}=\sum_{\nu }q_{\mu \nu }^{\left(j\right)}$ and the detection failure probabilities ${\bar{\xi }}_{e}=1-\xi_{e}$ and ${\bar{\xi }}_{i}=1-\xi_{i}$. In the second line, we use that $P(E_{\nu },M_{\mu },\mathrm{SC}|q,\lambda_{1},I)$ was already derived in Appendix 1 of [22]. The appearing integrals are of the type:(3)$$\begin{array}{c} \langle \lambda_{j}^{n}\rangle =\int_{0}^{\infty }d\lambda_{j}\lambda_{j}^{n}e^{-\lambda_{j}\left(1-{\bar{\xi }}_{e}{\bar{\xi }}_{i}\right)}p_{\Gamma }\left(\lambda_{j}|{\underline{\lambda }}_{j},\sigma_{j}\right) \end{array}$$
and solved in Appendix B. Note that the PDF describing the laser fluctuations enters the whole algorithm only within integrals of the above type. Thus, the theory can easily be adapted to different descriptions of the fluctuations by merely exchanging those integrals. The spectrum including false coincidences, ${\tilde{q}}_{\mu \nu }^{\left(1\right)}$, is then given by:(4)$$\begin{array}{cc} {\tilde{q}}_{\mu \nu }^{\left(1\right)} & =P\left(E_{\nu },M_{\mu }|\mathrm{SC},q^{\left(1\right)},\pi ,I\right)=\frac{P(E_{\nu },M_{\mu },\mathrm{SC}|q^{\left(1\right)},\pi ,I)}{\sum_{\mu \nu }P\left(E_{\nu },M_{\mu },\mathrm{SC}|q^{\left(1\right)},\pi ,I\right)}=\frac{q_{\mu \nu }^{\left(1\right)}+\kappa_{1}\;q_{\mu .}^{\left(1\right)}q_{.\nu }^{\left(1\right)}}{1+\kappa_{1}}\;. \end{array}$$
$\kappa_{1}$ turns out to be $\kappa_{1}={\bar{\xi }}_{e}\bar{\xi_{i}}\frac{\langle \lambda_{1}^{2}\rangle }{\langle \lambda_{1}^{1}\rangle }$. The false coincidences are represented by the term $\kappa_{1}\;q_{\mu .}^{\left(1\right)}q_{.\nu }^{\left(1\right)}$, and therefore $\kappa_{1}$ is a measure of the amount of false coincidences.

We now define ${\tilde{q}}^{\left(\beta \right)}$ analogously to ${\tilde{q}}^{\left(1\right)}$ and derive its dependence on $q^{\left(1\right)}$ and $q^{\left(2\right)}$, ${\underline{\lambda }}_{1}$ and ${\underline{\lambda }}_{2}$, and $\sigma_{1}$ and $\sigma_{2}$, abbreviated in the appendix as *q*, $\underline{\lambda }$, and σ. Therefore, we have to reassign the single coincidence event to the electron stemming from channel i∈{1,2} and the ion from channel j∈{1,2}, respectively.
$$\begin{array}{cc} {\tilde{q}}_{\mu \nu }^{\left(\beta \right)} & =\frac{\sum_{ij}P\left(E_{\nu },M_{\mu },{\mathrm{SC}}_{ij}|q^{},\pi ,I\right)}{\underset{ij}{\sum_{\mu \nu }}P\left(E_{\nu },M_{\mu },{\mathrm{SC}}_{ij}|q^{},\pi ,I\right)}\;. \end{array}$$
Distinguishing the case where the electron and ion are measured in coincidence and stem from the same (i=j) or a different (i≠j, or ¬i=j) channel, we obtain in Appendix C:(5)$$\begin{array}{cc} P(E_{\nu },M_{\mu },{\mathrm{SC}}_{ii}|q,\pi ,I) & =\xi_{e}\xi_{i}\left(q_{\mu \nu }^{\left(i\right)}\langle \lambda_{i}^{1}\rangle +{\bar{\xi }}_{e}{\bar{\xi }}_{i}q_{\mu .}^{\left(i\right)}q_{.\nu }^{\left(i\right)}\langle \lambda_{i}^{2}\rangle \right)\langle \lambda_{\neg i}^{0}\rangle \end{array}$$
(6)$$\begin{array}{cc} P(E_{\nu },M_{\mu },{\mathrm{SC}}_{i,\neg i}|q,\pi ,I) & =\xi_{e}\xi_{i}{\bar{\xi }}_{e}{\bar{\xi }}_{i}q_{.\nu }^{\left(i\right)}\langle \lambda_{i}^{1}\rangle q_{\mu .}^{\left(\neg i\right)}\langle \lambda_{\neg i}^{1}\rangle \;. \end{array}$$
In summary, we have:(7)$$\begin{array}{c} {\tilde{q}}_{\mu \nu }^{\left(\beta \right)}=\frac{q_{\mu \nu }^{\left(2\right)}+\kappa_{2}\left(q_{\mu .}^{\left(2\right)}q_{.\nu }^{\left(2\right)}+\alpha_{\nu }q_{\mu .}^{\left(2\right)}+\beta_{\mu }q_{.\nu }^{\left(2\right)}\right)+\gamma_{\mu \nu }}{Z_{2}} \end{array}$$
with the parameters:(8)$$\begin{array}{cccc} \alpha_{\nu } & =\left(\Omega -1\right)q_{.\nu }^{\left(1\right)}\;\; & \gamma_{\mu \nu } & =\frac{\langle \lambda_{1}^{1}\rangle \langle \lambda_{2}^{0}\rangle }{\langle \lambda_{1}^{0}\rangle \langle \lambda_{2}^{1}\rangle }\left(1+\kappa_{1}\right){\tilde{q}}_{\mu \nu }^{\left(1\right)}\\ \beta_{\mu } & =\left(\Omega -1\right)q_{\mu .}^{\left(1\right)}\;\; & Z_{2} & =1+\kappa_{2}\left(2\Omega -1\right)+\gamma_{..} \end{array}$$
where $\gamma_{..}=\sum_{\mu \nu }\gamma_{\mu \nu }$ and:$$\begin{array}{cccc} \kappa_{2} & ={\bar{\xi }}_{e}{\bar{\xi }}_{i}\frac{\langle \lambda_{2}^{2}\rangle }{\langle \lambda_{2}^{1}\rangle }\;\; & \Omega & =1+\frac{\langle \lambda_{1}^{1}\rangle \langle \lambda_{2}^{1}\rangle }{\langle \lambda_{1}^{0}\rangle \langle \lambda_{2}^{2}\rangle }\;. \end{array}$$
Up to now, we have defined all variables and dependencies we need for the derivation of the desired posterior distribution presented in the following sections.

### 3.2. The Posterior PDF

We use Bayes’ theorem to determine the posterior probability of the parameters we want to estimate, $\pi :=\{{\underline{\lambda }}_{1},\sigma_{1},{\underline{\lambda }}_{2},\sigma_{2},\xi_{i},\xi_{e}\}$, and the spectra $q^{\left(1\right)}=\left\{q_{\mu \nu }^{\left(1\right)}\right\}$ and $q^{\left(2\right)}=\left\{q_{\mu \nu }^{\left(2\right)}\right\}$.
(9)$$\begin{array}{c} \underset{\mathrm{Prior}}{\underbrace{p(q^{\left(1\right)},q^{\left(2\right)},\pi |I)}}\;\underset{\mathrm{Likelihood}}{\underbrace{P(D_{1},D_{2}|q^{\left(1\right)},q^{\left(2\right)},\pi ,I)}}=\underset{\mathrm{Evidence}}{\underbrace{P(D_{1},D_{2}|I)}}\;\underset{\mathrm{Posterior}}{\underbrace{p(q^{\left(1\right)},q^{\left(2\right)},\pi |D_{1},D_{2}I)}}\;, \end{array}$$
where capital *P* shall denote discrete and lower case *p* continuous distributions. $D_{1}$ and $D_{2}$ are two datasets, and $D_{1}$ contains the count rates $n^{\left(\alpha \right)}=\left\{n_{\mu \nu }^{\left(\alpha \right)}\right\}$ and $n^{\left(\beta \right)}=\left\{n_{\mu \nu }^{\left(\beta \right)}\right\}$. $\left\{n_{\mu \nu }^{\left(\rho \right)}\right\}$ counts how often the pair $\left(E_{\nu },M_{\mu }\right)$ was detected as a single coincidence event during the experiment ρ. Dataset $D_{2}$ contains $N_{N_{e},N_{i}}^{\left(\alpha \right)}$ and $N_{N_{e},N_{i}}^{\left(\beta \right)}$, which counts how many measurements lead to the detection of $N_{e}$ electrons and $N_{i}$ ions during the experiments α and β, respectively. In this case, it is expedient to use all detected events, not just single coincidences.

### 3.3. The Prior PDF

To define the prior PDF in Equation (Equation 9), we proceed with:$$\begin{array}{c} p\left(q^{\left(1\right)},q^{\left(2\right)},\pi |I\right)=p\left(q^{\left(1\right)},q^{\left(2\right)}|\pi ,I\right)\;p\left(\pi |I\right)\;. \end{array}$$
For the parameters’ prior, p(π|I), we use Jeffreys’ prior for scale variables for ${\underline{\lambda }}_{1}$ and ${\underline{\lambda }}_{2}$ and a flat prior for $\xi_{i}$, $\xi_{e}$, $\sigma_{1}$, and $\sigma_{2}$. The parameter ranges are ${\underline{\lambda }}_{1},{\underline{\lambda }}_{2},\sigma_{1},\sigma_{2}\in R^{+}$, and $0<\xi_{i},\xi_{e}<1$. The spectra are normalized according to Equation (Equation 1) and positive, $q_{\mu \nu }^{\left(1\right)},q_{\mu \nu }^{\left(2\right)}>0$. Our choice for the prior of the spectra,
(10)$$\begin{array}{c} p\left(q^{\left(1\right)},q^{\left(2\right)}|\pi ,I\right)=p\left(q^{\left(2\right)}|q^{\left(\alpha \right)},\pi ,I\right)\;p\left(q^{\left(1\right)}|\pi ,I\right)\;, \end{array}$$
are Dirichlet priors [30] in the spectra ${\tilde{q}}_{\mu \nu }^{\left(\alpha \right)}$ and ${\tilde{q}}_{\mu \nu }^{\left(\beta \right)}$,
$$\begin{array}{c} p\left({\tilde{q}}^{\left(\beta \right)}|q^{\left(\alpha \right)},\pi ,I\right)=\frac{1}{B(\left\{c_{\mu \nu }^{\left(\beta \right)}\right\})}\prod_{\mu \nu }{\tilde{q}}_{\mu \nu }^{\left(\beta \right)}{}^{c_{\mu \nu }^{\left(\beta \right)}-1}\;\delta \left(\tilde{S}-1\right)\;, \end{array}$$
with $\tilde{S}=\sum_{\mu \nu }{\tilde{q}}_{\mu \nu }^{\left(\rho \right)}$ and the normalization $B(\left\{c_{\mu \nu }^{\left(\rho \right)}\right\})$ being the multivariate beta function. We can always choose the prior to be uninformative (flat) by setting all $c_{\mu \nu }^{\left(\rho \right)}=1$. The choice of these Dirichlet priors is without loss of generality and justified by the likelihood being multinomial in these variables, as we will see later. Previously, with Equation (Equation 7), we found ${\tilde{q}}_{\mu \nu }^{\left(\beta \right)}$ to be a function of $q^{\left(\alpha \right)}$ and $q^{\left(2\right)}$, i.e., ${\tilde{q}}_{\mu \nu }^{\left(\beta \right)}={\tilde{Q}}^{\left(\beta \right)}\left(q^{\left(\alpha \right)},q^{\left(2\right)}\right)$. With the usual transformation rules, the first term in Equation (Equation 10) becomes:(11)$$\begin{array}{c} p\left(q^{\left(2\right)}|q^{\left(\alpha \right)},\pi ,I\right)=p\left({\tilde{q}}^{\left(\beta \right)}|q^{\left(\alpha \right)},\pi ,I\right)\times \left|\frac{d{\tilde{Q}}^{\left(\beta \right)}\left(q^{\left(\alpha \right)},q^{\left(2\right)}\right)}{dq^{\left(2\right)}}\right|\;. \end{array}$$
The appearing transformation of the Dirac distribution,
(12)$$\begin{array}{c} \delta \left(\tilde{S}-1\right)=\frac{Z_{2}}{1+2\kappa_{2}\Omega }\delta \left(S-1\right)\;, \end{array}$$
with $S=\sum_{\mu \nu }q_{\mu \nu }^{\left(2\right)}$, is derived in Appendix D, and the derivation of the Jacobian determinant,
(13)$$\begin{array}{c} \left|\frac{d{\tilde{Q}}^{\left(\beta \right)}\left(q^{\left(\alpha \right)},q^{\left(2\right)}\right)}{dq^{\left(2\right)}}\right|=\frac{{\left(1+\kappa_{2}\Omega \right)}^{N_{\mu }+N_{\nu }-2}\left(1+2\kappa_{2}\Omega \right)}{Z_{2}^{N_{\mu }N_{\nu }}}\;, \end{array}$$
is shown in Appendix E. $N_{\mu }$ and $N_{\nu }$ denote the total number of bins for ions (μ) and electrons (ν), respectively. Putting everything together, Equation (Equation 11) becomes: $$\begin{array}{c} p\left(q^{\left(2\right)}|q^{\left(\alpha \right)},\pi ,I\right)=\frac{{\left(1+\kappa_{2}\Omega \right)}^{N_{\mu }+N_{\nu }-2}\;\delta \left(\sum_{\mu \nu }q_{\mu \nu }^{\left(2\right)}-1\right)}{Z_{2}^{N_{\mu }N_{\nu }-1}B\left(\left\{c_{\mu \nu }^{\left(\beta \right)}\right\}\right)}\times \prod_{\mu \nu }{\left({\tilde{Q}}_{\mu \nu }^{\left(\beta \right)}\left(q^{\left(\alpha \right)},q^{\left(2\right)}\right)\right)}^{c_{\mu \nu }^{\left(\beta \right)}-1}\;. \end{array}$$
The second term in Equation (Equation 10), $p(q^{\left(1\right)}|\pi ,I)$, is obtained by setting Ω=1 and replacing experiment β by α and Channel 2 by 1; hence, the prior is fully determined.

### 3.4. The Likelihood

In this paper, we only use the single coincidence events $D_{1}$ for estimating the spectra $q^{\left(1\right)}$ and $q^{\left(2\right)}$ given the parameters π. This is justified by the fact that especially these events include relevant information about the spectrum, because the case of detecting more than one electron-ion pair does not allow one to link an electron to the ion it originates from, and the Bayesian approach would be different. The dataset $D_{2}$ will be used to determine the unknown parameters π. Therefore, the likelihood splits into:(14)$$\begin{array}{cc} P(D_{1},D_{2}|q^{\left(1\right)},q^{\left(2\right)},\pi ,I) & =P\left(D_{1}|q^{\left(1\right)},q^{\left(2\right)},\pi ,I\right)\;P\left(D_{2}|\pi ,I\right)\;. \end{array}$$
The first term separates into:$$\begin{array}{cc} P(D_{1}|q^{\left(1\right)},q^{\left(2\right)},\pi ,I) & =P\left(n^{\left(\alpha \right)}|q^{\left(1\right)},\pi ,I\right)P\left(n^{\left(\beta \right)}|q^{\left(1\right)},q^{\left(2\right)},\pi ,I\right)\;. \end{array}$$
Marginalizing over ${\tilde{q}}_{\mu \nu }^{\left(\alpha \right)}$ and ${\tilde{q}}_{\mu \nu }^{\left(\beta \right)}$ introduces the multinomial distribution:$$\begin{array}{c} P\left(n^{\left(\beta \right)}|{\tilde{q}}^{\left(\beta \right)},\pi ,I\right)=M\left(\left\{n^{\left(\beta \right)}\right\}|\left\{{\tilde{q}}^{\left(\beta \right)}\right\},\pi ,I\right) \end{array}$$
according to Appendix 2 of [22]. We now turn to the second term in Equation (Equation 14),
$$\begin{array}{cc} P(D_{2}|\pi ,I) & =P\left(\left\{N_{N_{e},N_{i}}^{\left(\alpha \right)}\right\}|\pi ,I\right)P\left(\left\{N_{N_{e},N_{i}}^{\left(\beta \right)}\right\}|\pi ,I\right)\;. \end{array}$$
With $N_{P}$ being the number of laser pulses; the terms are multinomial distributions, as well,
$$\begin{array}{c} P\left(\left\{N_{N_{e},N_{i}}^{\left(\rho \right)}\right\}|\pi ,I\right)=M\left(\left\{N_{N_{e},N_{i}}^{\left(\rho \right)}\right\}|\left\{{\tilde{P}}_{N_{e},N_{i}}^{\left(\rho \right)}\right\},N_{P}^{\left(\rho \right)}\right)\;. \end{array}$$
Since the PDF is the same for both experiments, α and β, we suppress the index ρ in the following considerations. The probability of detecting $N_{e}$ electrons and $N_{i}$ ions is:(15)$$\begin{array}{c} {\tilde{P}}_{N_{e},N_{i}}:=P\left(N_{e},N_{i}|\underline{\lambda },\sigma ,\xi_{e},\xi_{i},I\right)=\int_{0}^{\infty }d\lambda \;\underset{=:P_{N_{e},N_{i}}}{\underbrace{P(N_{e},N_{i}|\lambda ,\xi_{e},\xi_{i},I)}}p_{\Gamma }\left(\lambda |\underline{\lambda },\sigma \right)\;. \end{array}$$
$P_{N_{e},N_{i}}$ is already derived in Appendix 4 in [22] and has the general form:$$\begin{array}{c} P_{N_{e},N_{i}}=\sum_{n=\max(N_{e},N_{i})}^{N_{e}+N_{i}}a_{n}\left(\xi_{1},\xi_{i}\right)\lambda^{n}e^{-\lambda (1-{\bar{\xi }}_{e}{\bar{\xi }}_{i})}\;. \end{array}$$
Inserting in Equation (Equation 15) produces:$$\begin{array}{cc} {\tilde{P}}_{N_{e},N_{i}} & =\sum_{n=\max(N_{e},N_{i})}^{N_{e}+N_{i}}a_{n}\left(\xi_{e},\xi_{i}\right)\langle \lambda^{n}\rangle \end{array}$$
A list of ${\tilde{P}}_{N_{e},N_{i}}$ for $\max(N_{e},N_{i})\leq 3$ is given in Appendix F.

### 3.5. Remarks on the Posterior Sampling

In the previous sections, we fully determined the posterior $p(q^{\left(1\right)},q^{\left(2\right)},\pi |D_{1},D_{2},I)$. Ultimately, we are interested in the spectrum $q^{\left(2\right)}$, or rather say its probability $p(q^{\left(2\right)}|D_{1},D_{2},I)$, which can easily be achieved by integrating out $q^{\left(1\right)}$ and π in the posterior. For performing this integration and, more generally, computing expectation values, variances, and covariances of the posterior, we resort to numerical techniques, posterior sampling specifically. A suitable technique for sampling from this PDF is Markov Chain Monte Carlo (MCMC), which is based on a Markov chain that has the desired distribution as its equilibrium distribution. The technique is standard in Bayesian probability theory (see [30], especially Chapter 30 and the references therein). To generate the Markov chain, we used the Metropolis–Hastings algorithm with local updates in $q^{\left(1\right)}$, $q^{\left(2\right)}$, and π. We chose the step size for the update of each parameter separately to achieve an acceptance probability of ca. 50%. We discarded the first 20% of the Markov chain as means of thermalization and tested the convergence to the equilibrium distribution with several initial states. Correlations were checked with binning.

## 4. Mock Data Analysis

In this section, we demonstrate the performance of our algorithm including λ-fluctuations. There are two disturbing influences in the reconstruction of the spectra: false coincidences and pump-only background. We study these influences separately. First, we investigate the influence of the λ-fluctuations on the false coincidences. In the second part, we scrutinize our algorithm in the presence of a challenging pump-only background and λ-fluctuations.

### 4.1. False Coincidences

In the simulation, we have two different ion masses μ∈{p,f}, called parent (p) and fragment (f). We use the test spectra related to the ones reported in [22,26]. The electronic spectrum of the parent has a step-like form, and the electronic spectrum of the fragment consists of Gaussian peaks; see Figure 3. These challenging test spectra exhibit a strongly-varying parent/fragment-ratio as a function of the electron energy. Since we are firstly interested in the effect of λ-fluctuations on false coincidences, we suppress the background in our simulations (${\underline{\lambda }}_{1}\to 0$ and $\sigma_{1}\to 0$). Before studying the simulation results in detail, we investigate the influence of λ-fluctuations on false coincidences by analyzing Equation (Equation 7). Since we have only signal contributions ($q^{\left(2\right)}$), Equation (Equation 7) reduces to Equation (Equation 4) with the superscripts (2) and 2 instead of (1) and 1, respectively. The equation describes the connection between the measured spectrum ${\tilde{q}}^{\left(2\right)}$, including false coincidences, and the true spectrum $q^{\left(2\right)}$. $\kappa_{2}$ gives the weight between false and true coincidences,
(16)$$\begin{array}{c} \kappa_{2}={\bar{\xi }}_{e}\bar{\xi_{i}}\frac{\langle \lambda_{2}^{2}\rangle }{\langle \lambda_{2}^{1}\rangle }={\bar{\xi }}_{e}\bar{\xi_{i}}{\underline{\lambda }}_{2}\left\{\frac{1+\frac{\sigma_{2}^{2}}{{\underline{\lambda }}_{2}^{2}}}{1+\left(1-{\bar{\xi }}_{e}\bar{\xi_{i}}\right)\frac{\sigma_{2}^{2}}{{\underline{\lambda }}_{2}}}\right\}\;. \end{array}$$

This equation shows that the influence of λ-fluctuations on false coincidences is negligible in two cases: first, if the relative fluctuations $\sigma_{2}^{2}/{\underline{\lambda }}_{2}^{2}$ are small; and second, if the relation $\left(1-{\bar{\xi }}_{e}\bar{\xi_{i}}\right){\underline{\lambda }}_{2}\approx 1$ holds, the influence is negligible regardless of high relative fluctuations.

In our mock data analysis, we choose the parameters $\xi_{i}=\xi_{e}=0.5$ and $N_{p}=10^{7}$, $\sigma_{2}=0.5$, and ${\underline{\lambda }}_{2}=\left\{1.5,0.5\right\}$. Now, we turn to the parameter estimates summarized in Table 1. The algorithm reported in [22], which does not include λ-fluctuations, provides estimates for $\lambda_{2}$, $\xi_{i}$, and $\xi_{e}$ with about 5% (for ${\underline{\lambda }}_{2}=1.5$) and 13% (for ${\underline{\lambda }}_{2}=0.5$) relative deviation from the real values. The differences are caused by the assumption of having a constant $\lambda_{2}$ without any statistical noise, which is simply wrong, since we generated our mock data by drawing $\lambda_{2}$ from the Γ-distribution with the parameters ${\underline{\lambda }}_{2}$ and $\sigma_{2}$. As expected, $\lambda_{2}$ estimated by the algorithm ignoring λ-fluctuations lies within ${\underline{\lambda }}_{2}\pm \sigma_{2}$. The algorithm reported in this paper includes the λ-fluctuations and is therefore able to give reliable estimates for all parameters.

We will now show that even if the parameter estimation of the algorithm not including λ-fluctuations is slightly off, there is only a small difference in the reconstructed spectra. While at ${\underline{\lambda }}_{2}=1.5$, λ-fluctuations do not affect the reconstruction (Figure 3a,b), at ${\underline{\lambda }}_{2}=0.5$, the algorithm including λ-fluctuations (green line) is slightly closer to the true spectrum than the algorithm not including λ-fluctuations (blue line); see Figure 3c,d. In this parameter regime, neglecting λ-fluctuations leads to $\kappa_{2}=0.125$, while Equation (Equation 16) produces $\kappa_{2}=0.182$. Therefore, the algorithm not including λ-fluctuations underestimates the amount of false coincidences.

Altogether, the influence of λ-fluctuations on false coincidences in the reconstructed spectra is small in the experimentally-relevant parameter regime, which is in accordance with Mikosch et al. [26]. Still, including λ-fluctuations is important in the parameter estimation.

### 4.2. Background Subtraction

To analyze the influence of λ-fluctuations on the background subtraction, we use the same spectra as in Section 4.1. Now, the step-like spectrum is chosen as the signal and the spectrum consisting of Gaussian peaks as the background. We restrict ourselves to one ion mass to exclude the influence of false coincidences stemming from two ion masses. If we neglect false coincidences, the prefactor $\frac{\langle \lambda_{1}^{1}\rangle \langle \lambda_{2}^{0}\rangle }{\langle \lambda_{1}^{0}\rangle \langle \lambda_{2}^{1}\rangle }$ in $\gamma_{\mu \nu }$ defined in Equation (Equation 8) is:$$\begin{array}{c} \frac{\langle \lambda_{1}^{1}\rangle \langle \lambda_{2}^{0}\rangle }{\langle \lambda_{1}^{0}\rangle \langle \lambda_{2}^{1}\rangle }=\frac{{\underline{\lambda }}_{1}}{{\underline{\lambda }}_{2}}\left\{\frac{1+\left(1-{\bar{\xi }}_{e}\bar{\xi_{i}}\right)\frac{\sigma_{2}^{2}}{{\underline{\lambda }}_{2}}}{1+\left(1-{\bar{\xi }}_{e}\bar{\xi_{i}}\right)\frac{\sigma_{1}^{2}}{{\underline{\lambda }}_{1}}}\right\}\;. \end{array}$$
This factor represents the ratio between the background (1) and signal (2). The equation shows that the corrections cancel if ${\underline{\lambda }}_{1}={\underline{\lambda }}_{2}$ and $\sigma_{1}=\sigma_{2}$; see Figure 4a,b. If $\sigma_{1}^{2}/{\underline{\lambda }}_{1}<\sigma_{2}^{2}/{\underline{\lambda }}_{2}$ (Figure 4c), the algorithm without λ-fluctuations (blue line) underestimates the background, leading to peaks in every step. In the opposite case, $\sigma_{1}^{2}/{\underline{\lambda }}_{1}>\sigma_{2}^{2}/{\underline{\lambda }}_{2}$ (Figure 4d), the algorithm without λ-fluctuations (blue line) overestimates the background, leading to notches in every step. The algorithm including λ-fluctuations (green line) is able to reconstruct the signal correctly in all scenarios.

Similar to the mock data analysis in the previous section, not including λ-fluctuations leads to deviations in the parameter estimation in all of the simulated test cases; see Table 2. As expected, $\lambda_{j}$ estimated by the algorithm ignoring λ-fluctuations lies within ${\underline{\lambda }}_{j}\pm \sigma_{j}$. Obviously, including λ-fluctuations in the parameter estimation becomes important especially for high $\sigma_{1}$ or $\sigma_{2}$. In summary, the influence of λ-fluctuations on the background subtraction is important if the magnitude of the relative λ-fluctuations of one channel is large and does not compare to the other.

## 5. Application to Experimental Data

With our setup for time-resolved PEPICO studies on gas phase molecules (see Figure 2), we were not able to produce stable λ-fluctuations with a sufficiently large σ to demonstrate a systematic influence on the reconstructed spectra. The output of our commercial laser system (Coherent Vitara oscillator and Legend Elite Duo amplifier) turned out to be too stable for this purpose. We tried to simulate a less stable setup by operating the Optical Parametric Amplifier (OPA) at intensities below the saturation threshold, which induces λ-fluctuations. In this setting, however, the pulse energy also undergoes temporal drifts on time scales of the measurement, which are different from the statistical fluctuations and not accounted for in our analysis as a setup would not be operated in this regime.

Nevertheless, by using the algorithm including the λ-fluctuations, we find that the fluctuations were in the order of σ=O(0.001) in the experimentally-relevant parameter regime. The parameters estimated by the different algorithms are depicted in Table 3. Finally, we note that less stable systems are expected to suffer from significant λ-fluctuations, which have to be accounted for in the reconstruction process of the spectrum. This will in particular be the case if multiple nonlinear optical processes like in the OPA are used in the pump path of a pump-probe setup. The fluctuations get even more prominent if multiphoton processes are used for the excitation process, because these processes depend nonlinearly on the pulse power.

## 6. Conclusions

We used Bayesian probability theory to analyze data obtained from pump-probe photoionization experiments with photoelectron-photoion coincidence detection. We extended the algorithm developed previously in [22] by including fluctuations of the laser intensity as a random variable λ. Based on challenging mock data, we have demonstrated the reliability of the developed algorithm. In accordance with Mikosch et al. [26], the influence of λ-fluctuations on false coincidences was small in a certain parameter regime. We derived the condition $\left(1-{\bar{\xi }}_{e}\bar{\xi_{i}}\right)\underline{\lambda }\approx 1$ for negligible influence of λ-fluctuations on false coincidences, even at high relative fluctuations. ${\bar{\xi }}_{e}$ and ${\bar{\xi }}_{i}$ denote the complement of the detection probabilities of electrons ($\xi_{e}$) and ions ($\xi_{i}$), respectively. In the case of the pump-only background (Channel 1) and the signal (Channel 2) contained in the pump-probe measurements, neglecting λ-fluctuations underestimates the background signal in the case $\sigma_{1}^{2}/{\underline{\lambda }}_{1}<\sigma_{2}^{2}/{\underline{\lambda }}_{2}$. The other way around, the background signal is overestimated. In both scenarios, including λ-fluctuations is important for estimating the parameters ${\underline{\lambda }}_{1}$, ${\underline{\lambda }}_{2}$, $\sigma_{1}$, $\sigma_{2}$, $\xi_{i}$, and $\xi_{e}$ correctly. In our application to the experimental data, we find that the relative laser fluctuations in the experimental setup are fortunately too small to see effects on the reconstructed spectra. With the developed algorithm, we were able to determine the experimental λ-fluctuations to be in the order of σ=O(0.001). Compared to conventional subtraction of the pump-only spectrum from the pump-probe spectrum, the Bayesian approach provides several important advantages: (i) It results in a significant increase of the signal-to-noise ratio. (ii) It does not overestimate the pump-only contribution and never leads to negative spectra because the relative weight of the pump-only contribution is self-consistently determined. (iii) Spectral signatures based on false coincidences are eliminated, allowing for higher signal rates. (iv) It includes consistently all prior knowledge, such as positivity, and (v) a confidence interval is obtained for the estimated spectrum. (vi) It is applicable to any experimental situation and noise statistics, as demonstrated in this paper for the case of λ-fluctuations.

## Figures and Tables

**Figure 1 entropy-21-00093-f001:**
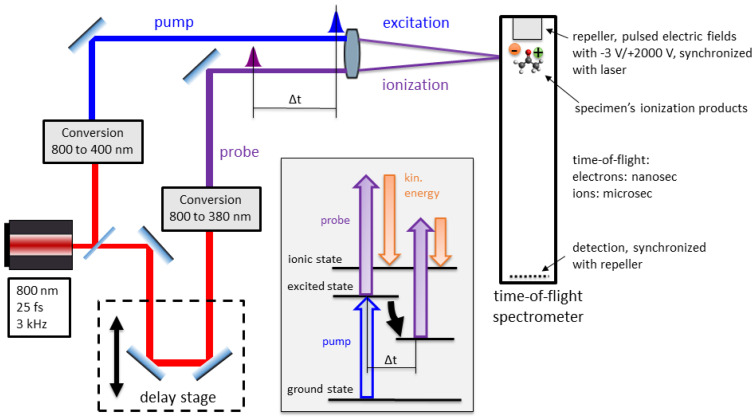
Utterly simplified sketch of a time-resolved photoionization study carried out with a pump-probe setup and a time-of-flight spectrometer. A commercial Ti:sapphire laser system delivers pulses of 800 nm in center wavelength and 25 fs in temporal length at a repetition rate of 3 kHz. The delay stage is used to control the length of the optical path, and hence the time delay. The energy level diagram shows how the electron kinetic energy, given the energy of the states and the photons, identifies the state the system was in at the moment of ionization. A detailed description of the setup can be found in our previous publications [7,28].

**Figure 2 entropy-21-00093-f002:**
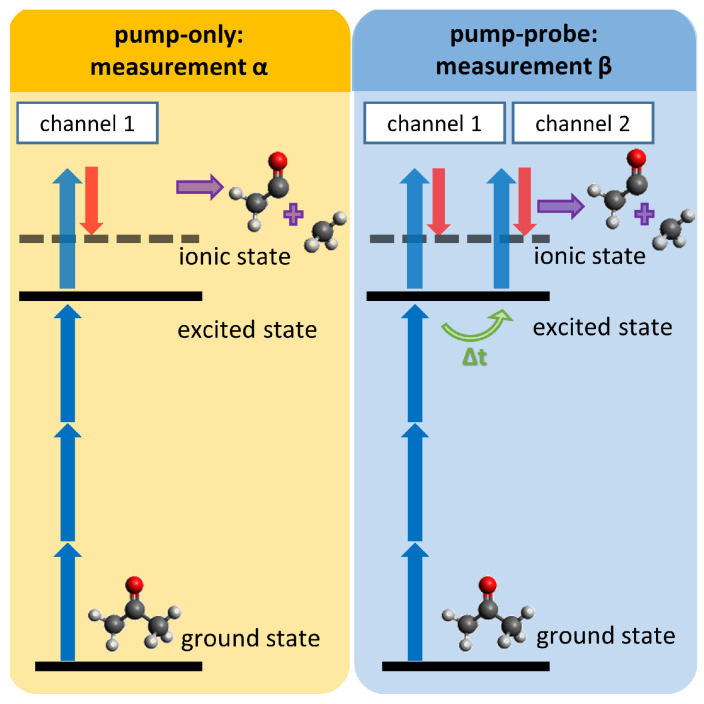
Pump-probe ionization scheme to investigate excited state dynamics in molecules.

**Figure 3 entropy-21-00093-f003:**
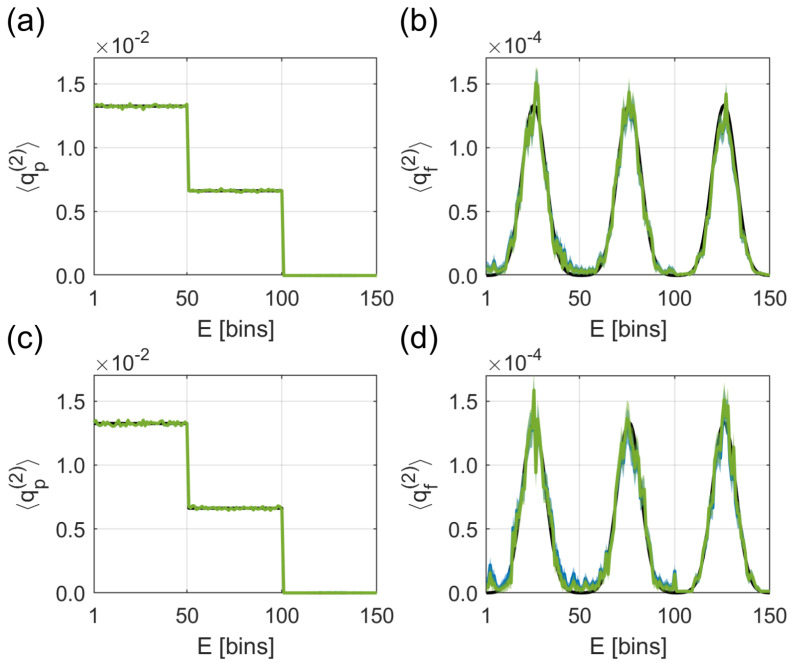
Simulation with mock data for studying the influence of λ-fluctuations on false coincidences. The black lines are the spectra used to generate the data; the green (blue) lines including ±σ error bands are the reconstructed spectra (not) including λ-fluctuations in the reconstruction. The parameters are $\xi_{i}=\xi_{e}=0.5$ and $N_{p}=10^{7}$. For ${\underline{\lambda }}_{2}=1.5$, differences between the algorithms are negligible even at relatively high λ-fluctuations with $\sigma_{2}=0.5$; see spectra (**a**,**b**). When choosing ${\underline{\lambda }}_{2}=0.5$ (**c**,**d**), the algorithm not including λ-fluctuations produces small deviations, e.g., underestimation of the false coincidences at the first Gaussian in the fragment spectrum.

**Figure 4 entropy-21-00093-f004:**
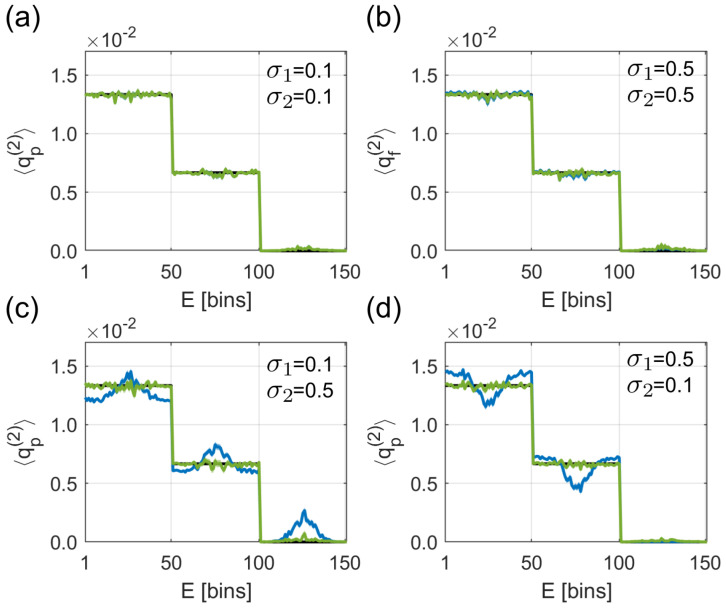
Simulated test spectra for studying the influence of λ-fluctuations on the background subtraction. The parameters are ${\underline{\lambda }}_{1}={\underline{\lambda }}_{2}=0.5$, $\xi_{i}=\xi_{e}=0.5$, and $N_{p}=10^{7}$. $\sigma_{1}$ and $\sigma_{2}$ are different for every sub-figure. If $\sigma_{1}=\sigma_{2}=0.1$ (**a**) or $\sigma_{1}=\sigma_{2}=0.5$ (**b**), both algorithms (with (green line) and without (blue line) including λ-fluctuations) reconstruct the spectra correctly. $\sigma_{1}=0.1$ and $\sigma_{2}=0.5$ lead to an underestimation of the background when neglecting λ-fluctuations (**c**). Overestimation of the background happens in the case of $\sigma_{1}=0.5$ and $\sigma_{2}=0.1$ (**d**).

**Table 1 entropy-21-00093-t001:** Estimated parameters ${\underline{\lambda }}_{2}$, $\sigma_{2}$, $\xi_{i}$, and $\xi_{e}$. In the lines showing the results of the algorithm presented in [22], $\lambda_{2}$ is shown instead of ${\underline{\lambda }}_{2}$. Each value denotes the mean and standard deviation of the parameter’s distribution.

	*λ*${}_{2}$	$\sigma_{2}$	$\xi_{i}$	$\xi_{e}$
Parameters (Figure 3a,b)	1.5	0.5	0.5	0.5
Algorithm in [22]	1.4177±0.0008	-	0.5235±0.0003	0.5235±0.0003
Algorithm with λ-fluctuations	1.499±0.001	0.501±0.002	0.4998±0.0003	0.5000±0.0003
Parameters (Figure 3c,d)	0.5	0.5	0.5	0.5
Algorithm in [22]	0.4365±0.0003	-	0.5618±0.0004	0.5621±0.0004
Algorithm with λ-fluctuations	0.4992±0.0005	0.499±0.001	0.5000±0.0004	0.5004±0.0004

**Table 2 entropy-21-00093-t002:** Estimated parameters ${\underline{\lambda }}_{1}$, ${\underline{\lambda }}_{2}$, $\sigma_{1}$, $\sigma_{2}$, $\xi_{i}$, and $\xi_{e}$. The parameter regimes denoted by the identifications (a–d) are according to Figure 4. For each parameter set, the first line denotes the true value, while Line 2 (3) contains the parameter estimation performed with the algorithm without (with) λ-fluctuations, respectively. In the lines showing the results of the algorithm ignoring λ-fluctuations, $\lambda_{j}$ is shown instead of ${\underline{\lambda }}_{j}$. Each value denotes the mean and standard deviation of the parameter’s distribution.

	*λ*${}_{1}$	*λ*${}_{2}$	$\sigma_{1}$	$\sigma_{2}$	$\xi_{i}$	$\xi_{e}$
(a)	0.5	0.5	0.1	0.1	0.5	0.5
	0.4970±0.0003	0.4972±0.0005	-	-	0.5032±0.0002	0.5029±0.0002
	0.5000±0.0003	0.5007±0.0005	0.098±0.002	0.101±0.005	0.5003±0.0003	0.5000±0.0003
(b)	0.5	0.5	0.5	0.5	0.5	0.5
	0.4367±0.0003	0.4317±0.0004	-	-	0.5618±0.0002	0.5620±0.0002
	0.5002±0.0004	0.5009±0.0006	0.5026±0.0008	0.496±0.002	0.4996±0.0003	0.4998±0.0003
(c)	0.5	0.5	0.1	0.5	0.5	0.5
	0.4840±0.0003	0.4608±0.0005	-	-	0.5201±0.0002	0.5201±0.0002
	0.4991±0.0003	0.5013±0.0006	0.098±0.002	0.500±0.002	0.5001±0.0002	0.5001±0.0003
(d)	0.5	0.5	0.5	0.1	0.5	0.5
	0.4452±0.0003	0.4675±0.0004	-	-	0.5441±0.0002	0.5440±0.0002
	0.4998±0.0004	0.5005±0.0006	0.5002±0.0009	0.102±0.004	0.4998±0.0003	0.4997±0.0003

**Table 3 entropy-21-00093-t003:** Estimated parameters ${\underline{\lambda }}_{1}$, ${\underline{\lambda }}_{2}$, $\sigma_{1}$, $\sigma_{2}$, $\xi_{i}$, and $\xi_{e}$. Line 1 (2) contains the parameter estimations performed with the algorithm without (with) λ-fluctuations, respectively. In the line showing the results of the algorithm ignoring λ-fluctuations, $\lambda_{j}$ is shown instead of ${\underline{\lambda }}_{j}$. Each value denotes the mean and standard deviation of the parameter’s distribution.

*λ*${}_{1}$	*λ*${}_{2}$	$\sigma_{1}$	$\sigma_{2}$	$\xi_{i}$	$\xi_{e}$
0.3328±0.0009	0.132±0.001	-	-	0.3247±0.0008	0.542±0.001
0.3328±0.0009	0.1335±0.0003	0.0004±0.0010	0.0004±0.0012	0.3245±0.0008	0.541±0.001

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
