# Peer review of "Bayesian Analysis of Femtosecond Pump-Probe Photoelectron-Photoion Coincidence Spectra with Fluctuating Laser Intensities"

_entropy, 2019, doi:10.3390/e21010093_

Round 1

Reviewer 1 Report

I have already heard this work presented, and the manuscript confirms that high standard.  I only have a few stylistic suggestions for improvement.

1: The array of probability formulas is rather forbidding, and I recommend a clearer separation of the Bayesian family of prior, likelihood, evidence and posterior.  Indeed an explicit preliminary statement of Bayes' Theorem

                           Prior x Likelihood = Evidence x Posterior

would confirm the framework and focus the presentation.  In particular, the fundamental prior probability distribution isn't currently defined.  Readers can only assume (from manuscript line 142) that the relevant parameters (underline_lambda, sigma, etc.) to be estimated are being assigned uniform distributions over whatever ranges are adequate to support the posterior.  Please explain and define the prior model *before* introducing mock data and getting to the likelihood function and then proceeding to the Bayesian computation.

2. Description in the manuscript proper of the actual computation is disconcertingly sparse, being little more than reference to the tome by von der Linden et.al. [30]. Where in [30] should readers look?  I admit that appropriate detail is given in the supplementary material, but I suggest putting a few sentences into the manuscript, if only to confirm the methodology for readers who already know about MCMC.

3. Reading through the PEPICOBayes manual in supplementary material, I found myself unable to understand the sentence "Note that the standard deviation of for the step distribution for $\sigma_j$ is $\min(\sigma_{\sigma_j}, \sigma_j)".  I'm not sure if its the "of for" grammar or the double subscripting of the symbols that's confusing, but something seems to be awry and could do with minor editing.

John Skilling

Author Response

See attached pdf

Reviewer 2 Report

The paper presents Bayesian modelling for analyzing PEPICO experiments. The application is explained, a model proposed and fit to data.

I am not a specialist of the application, so I would assume that the other referee should know whether the approach of the authors is novel or not. However, what is clear is the simplistic presentation and discussion in Section 3. All the details here are exceptionally basic and the presentation could be half as long with much more sophisticated presentation. I would like the authors to write down the posterior they want to make inference from and why it makes sense for the data. Parameterizations of gamma distributions and integrating out variables are not appropriate for an international journal and should not be present (I am more than willing to concede (and check if I want) that the authors can do these things. Also, why are all the displayed equations numbered, without being referred to in the text? This should not be done.

Author Response

See attached pdf

Round 2

Reviewer 1 Report

Yes, I'm happier now.

Typo at top of page 6, where "marginal" is written "martingal".  That could be confusing because "martingale" is a technical term in statistics of no relevance here.

Reviewer 2 Report

The changes are fine.